# Port Structure Inspection Based on 6-DOF Displacement Estimation Combined with Homography Formulation and Genetic Algorithm

**Jiyoung Min [1], Yuseok Bang [2], Hyuntae Bang [3] and Haemin Jeon [3],***

[1] Department of Structural Engineering Research, Korea Institute of Civil Engineering and Building Technology, Goyang-si 10233, Korea; amote83@kict.re.kr
[2] SRH Solution, Hwaseong-si 18496, Korea; richardbang.srh@gmail.com
[3] Department of Civil and Environmental Engineering, Hanbat National University, Daejeon 34158, Korea; htbang@hanbat.ac.kr
* Correspondence: hjeon@hanbat.ac.kr; Tel.: +82-42-821-1103

**Abstract:** A vision sensor-based 6-DOF displacement evaluation method incorporating a genetic algorithm was proposed to monitor the critical defects of port infrastructure, such as deflection, slope, and slip. The 6-DOF behavior of the port structure, including subsidence, was estimated based on the specification of the target and fixed structures nearby. The method calculates the relative position of the target port structure and measures the movement of the structure over time. To improve the measurement accuracy, a genetic algorithm was used to adjust the intrinsic parameters that were previously estimated using the checkerboards. The results of measuring 6-DOF displacements based on the tuned intrinsic parameters confirmed that it has the potential to accurately measure the 6-DOF behavior of port facilities. The possibility of field application was examined through an artificial movement that was induced in the image of the port facility to create an arbitrary displacement between two points.

**Keywords:** port structure; displacement measurement; genetic algorithm; homography estimation

## 1. Introduction

The aging and deterioration of port facilities in the Republic of Korea has become an issue that should be addressed. As of 2020, 49.4% (538 locations) of port facilities are aged over 20 years, while 13.1% (143 locations) are aged over 40 years. According to the safety inspection and precision safety diagnosis reports of port facilities, in the facilities of more than 20 years age, the A-grade ratio decreased sharply, while in the case of 40 years or more, the A- and B-grade ratios tended to decrease. Moreover, the increase in the intensity and frequency of natural disasters related to climate change increase the variability of the design external force and enhance the possibility of large-scale damage to aging port facilities [1]. Figure 1 shows the critical damage cases that have occurred in port facilities. In response, the Ministry of Oceans and Fisheries of the Republic of Korea has established a national roadmap in 2020 for the smart sensing, monitoring, analysis, evaluation, and repair of port facilities for proactive and timely maintenance. In the detailed guidelines for infrastructure safety inspection and precision safety diagnosis, the critical major defects in port facilities are defined as: foundation scour, damage and corrosion of piles, loss of internal force due to carbonation and chloride attack in concrete, corrosion of lock gate facilities, and the normal displacement and settlement of berthing structures [2,3].

The settlement and normal displacement of berthing structures is generally evaluated by surface level surveying; the foundation scour should be evaluated by divers, and the members towards the sea should be inspected by inspectors moving in a boat. These evaluations and inspections are carried out every few years, and thus continuous monitoring is difficult. Attachment of various electric sensors is one method to monitor the behavior,

such as displacement, settlement, slip, and slope, but it is complicated to organize the sensing system with consideration of the berth, salt attack, high-risk work on the members towards the sea, and facility users' route [4–7]. Thus, in this paper, we present a technique for measuring the precise behavior of a berthing structure that could be caused by scouring, settlement, slip, damage, material deterioration, and so forth.

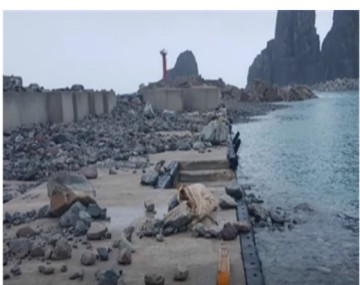 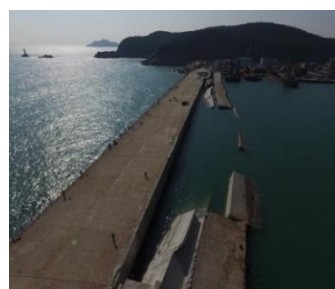 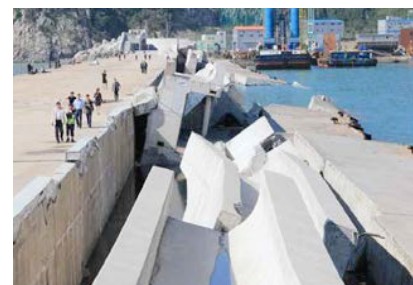

**Figure 1.** Critical damage to the facilities.

The behavior of the structures using a vision-based non-contact type displacement measurement system has gained rapid developments in the past decade [8]. Kohut et al. (2013) presented a vision-based deflection measurement method using the digital image correlation coefficient [9]. Jeon et al. (2014) proposed a 6-DOF translational and rotational displacement measurement system with a vision sensor and a uniquely designed marker [10]. Ye et al. (2015) proposed a multi-point displacement measurement method by use of a pattern-matching algorithm [11]. Feng et al. (2015) proposed a structural displacement measurement method with a subpixel resolution using the upsampled cross correlation algorithm [12]. Zhou et al. (2020) proposed a videogrammetric technique for displacement monitoring that eliminates the measurement error due to the image drift induced by temperature variation [13]. Most of the aforementioned non-contact type vision-based displacement measurement systems, however, have one of following drawbacks: only estimated deflection, which is 1-DOF displacement, markers are attached on the structures for feature points detection, or the accuracy of the measurements highly depends on the camera calibration results for calculating intrinsic parameters.

The 6-DOF displacement also can be measured using structured light composed of lasers and vision sensors [14–16]. The translational and rotation displacement measurement system called a paired structured light system composed of two sides facing each other, each with one or two lasers, a screen, and a camera. The lasers on each side project their beams on the screen on the opposite side, and a camera near the screen captures an image of the screen. By calculating the positions of the laser beams, the relative displacement between two sides can be estimated. In a follow-up study conducted by the same research group, a 2-DOF manipulator was introduced to an increased range of the displacement measurement. In the case of a visually servoed paired structure light system, the displacement can be estimated with an error within 0.2 mm and 0.2 deg, but the installation of a relatively heavy sensing system on port structures and mobile platforms is required. Therefore, in this paper, a displacement measurement method using the fixed intrinsic parameter of the camera is applied to measure the displacement of 6-DOF between the camera and the fixed/port structure; with its use, the movement of the port structure based on fixed structure can be measured. In this paper, the floating port structure was assumed to be a rigid body, and it was assumed that there was no deformation in shape. Since the displacement estimation of the structure is highly dependent on the camera-intrinsic parameter, in this paper, the intrinsic parameter is tuned based on the given measured translational and rotational displacements using a genetic algorithm. An indoor model experiment and an outdoor field image-based experiment were performed, and the results of the experiments confirmed that translational and rotational displacements are estimated more precisely

after calibrating the intrinsic parameters of the vision sensor, and the proposed technique is applicable to the field.

The remainder of the paper is organized as follows. In Section 2, the translational and rotational displacement estimation method using a vision sensor is described. The application of the genetic algorithm for tuning the camera-intrinsic parameters is introduced in Section 3. To validate the performance and applicability of the proposed method, the experimental tests using model structures and captured image with a drone are conducted and the results are discussed in Section 4. Conclusions and further research directions are discussed in Section 5.

## 2. 6-DOF Displacement Estimation Using Vision Sensor

The 6-DOF relative displacements that include translational and rotational displacements in three axes can be estimated by using positions of feature points in world coordinates and the intrinsic parameters of a vision sensor. The intrinsic parameters determine the optical properties of the camera lens, including the focal lengths, principal points, and distortion coefficients. Figure 2 represents the geometry view of the feature points in both the world and image planes. In the figure, $Q_i$ and $q_i$ ($i = 1, \ldots, N$) denote the corresponding points of the world and image planes, respectively, where $N$ is the number of feature points. The points in the world plane, $Q_i$, defined as $Q_i = [X\ Y\ Z\ 1]^T$, are represented in the three-dimensional coordinate system. The corresponding points, $q_i$, defined as $q_i = [u\ v\ 1]^T$, are represented in two-dimensional space. The relationship between the two planes can be expressed in terms of matrix multiplication, as follows:

$$\begin{bmatrix} u \\ v \\ 1 \end{bmatrix} = \begin{bmatrix} f_u & 0 & c_u \\ 0 & f_v & c_v \\ 0 & 0 & 1 \end{bmatrix} \begin{bmatrix} x_d \\ y_d \\ 1 \end{bmatrix}, \tag{1}$$

$$\begin{bmatrix} x_d \\ y_d \end{bmatrix} = \begin{bmatrix} x(1 + K_1 r^2 + K_2 r^4) + 2K_3 xy + K_4(r^2 + 2x^2) \\ y(1 + K_1 r^2 + K_2 r^4) + 2K_3(r^2 + 2xy^2) + K_4 xy \end{bmatrix} \tag{2}$$

$$\begin{bmatrix} X_c \\ Y_c \\ Z_c \end{bmatrix} = \begin{bmatrix} r_{11} & r_{12} & r_{13} & t_x \\ r_{21} & r_{22} & r_{23} & t_y \\ r_{31} & r_{32} & r_{33} & t_z \end{bmatrix} \begin{bmatrix} X \\ Y \\ Z \\ 1 \end{bmatrix} \tag{3}$$

where $f_u$ and $f_v$ are the focal length, $c_u$ and $c_v$ represent the principal point where the focal axis of the camera intersects the image plane; $K_1$ and $K_2$ are the radial distortion coefficients, $K_3$ and $K_4$ are the tangential distortion coefficients; $r$ and $t$ are parameters of the rotation matrix and translation vector. In Equation (2), $x = X_c/Z_c$, $y = Y_c/Z_c$, and $r^2 = x^2 + y^2$, where $X_c$, $Y_c$, and $Z_c$ are defined in Equation (3). The homography matrix composed of intrinsic and extrinsic camera parameters explains how to map pixels on a 2D image to the corresponding real-world coordinates in 3D scenes, as shown in Equations (1)–(3) [17,18]. By using the feature points on the same level, $3 \times 3$ sized homography matrix can be used with the given 2D-to-2D point correspondences. Since the degree of freedom of the homography matrix is equal to eight, at least four point-to-point correspondences are required. In other words, the rotation matrix and the translation vector can be obtained with the known positions of more than four feature points ($N \geq 4$) [18]. By calculating rotational and translational displacements from the vision sensor to the target and the fixed structures, the relative 6-DOF displacement of the target structure can be estimated.

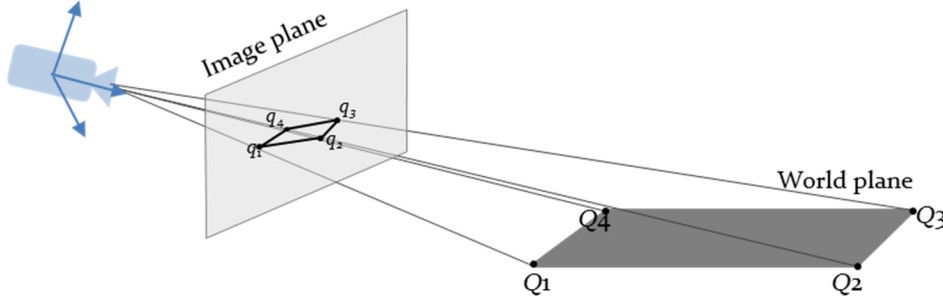

**Figure 2.** Homogeneous transformation between world and image planes.

Figure 3 shows the entire process of the relative displacement estimation between the two structures using image processing techniques. The figure shows that the camera captures the image of the structures, then the camera lens distortion is corrected by using the previously calculated intrinsic parameters. From the undistorted image, the feature points of the structures, including corners, are detected by using various image processing techniques, such as binarization, and corner detection at the sub-pixel level. By calculating at least four feature point positions, 6-DOF displacement between the camera and the structures can be estimated. The relative displacement can be estimated by using the previously calculated displacements on each structure. The relative displacement between two structures, the fixed and target structures, can be estimated using the following equations:

$$
\begin{aligned}
{}^F D_T(x,y,z,\theta,\varphi,\psi) &= T(x,y,z)R_x(\theta)R_y(\varphi)R_z(\psi) \\
&= \begin{bmatrix}
c_\varphi c_\psi & -c_\varphi s_\psi & s_\varphi & x \\
s_\theta s_\varphi c_\psi + c_\theta s_\psi & -s_\theta s_\varphi s_\psi + c_\theta s_\psi & s_\theta c_\varphi & y \\
-c_\theta s_\varphi c_\psi + s_\theta s_\psi & c_\theta s_\varphi s_\psi + s_\theta s_\psi & c_\theta c_\varphi & z \\
0 & 0 & 0 & 1
\end{bmatrix}
\end{aligned}
\tag{4}
$$

$$
{}^F D_T = {}^F D_C \cdot {}^C D_T
\tag{5}
$$

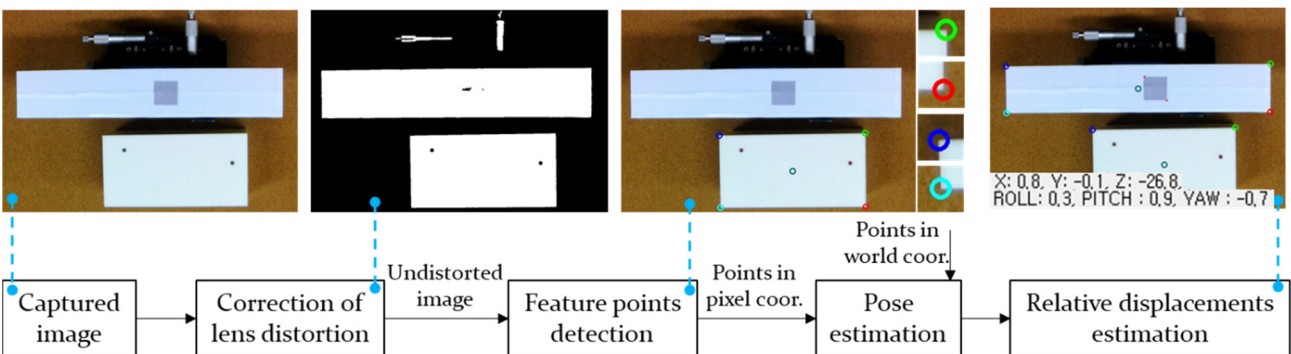

**Figure 3.** A block diagram of the displacement estimation process.

In Equation (4), ${}^F D_T$ is the transformation matrix composed of the 6-DOF relative displacement between fixed coordinate relative to the target coordinate, and *F* and *T* indicate the fixed and target structures, respectively. The matrix consists of the product of translation matrix *T(x,y,z)* along *X*, *Y*, and *Z* axes with rotation matrices $R_x(\theta)$, $R_y(\varphi)$, and $R_z(\psi)$ about *X*, *Y*, and *Z* axes, respectively. In the equation, $S_\theta$ and $C_\theta$ denote $\sin\theta$ and $\cos\theta$, respectively. The details of each matrix can be found in [19]. In Equation (5), ${}^F Dc$ and ${}^T Dc$ are the relative displacements between fixed or target coordinates relative to the camera coordinates, indicated as *C*. ${}^C D_T$ can be estimated by inverting ${}^T D_C$. The

relative displacements estimated from images taken at different time $t-1$ and $t$ are used to estimate the structural behavior, as follows:

$$^{T,t}D_{T,t-1} = {}^{T}D_{F,t} \cdot {}^{F}D_{T,t-1} \tag{6}$$

where ${}^{T}D_{F,t}$ equals the translational and rotational displacements between the fixed and target structures at time $t$.

### 3. Application of Genetic Algorithm for Optimization of the Camera-Intrinsic Parameters

Metaheuristic algorithms has been developing rapidly in recent years to solve real-life complex problems in various fields [20,21]. Most of the metaheuristic algorithms are inspired from biological evolution, swarm behavior, and laws of physics and can be classified into two categories such as single solution and population-based metaheuristics [22]. In comparison with the single solution approach that improve the solution by using local search, the population-based metaheuristics maintain the diversity in the population and avoid sucking in local optima [23]. Among the population-based metaheuristic algorithms, genetic algorithm (GA), which is one of the well-known algorithms, is used to find the parameter sets in the homography equation. GA, introduced by A. S. Fraser in 1957, is guaranteed to converge to an optimal solution in multivariable function by repeating population generation, fitness/penalty evaluation, selection, reproduction, crossover, and mutation [24,25]. Compared to other optimization methods, it is capable of solving any optimization problem based on a chromosome approach, and of handling a multiple solution search space with less complexity, and in a more straightforward manner [26]. GA is widely used in various research fields due to its advantage in creating models in a probabilistic manner. It includes new information in a non-arbitrary way, despite the limitation of being time-consuming and computationally intensive.

Algorithm 1 shows the entire procedure of optimizing the intrinsic parameters of the homography equation by using GA. The algorithm shows that the initial population of chromosomes, composed of parameters of the homography equation, such as $P_{set} = [f_u, f_v, c_u, c_v, \mathbf{K}]$, where $\mathbf{K}$ includes the radial distortion coefficients ($K_1$ and $K_2$), and tangential distortion coefficients ($K_3$ and $K_4$) is generated. After the generation, the penalty of each chromosome is evaluated, and the best chromosome is obtained that minimizes the difference between the estimated and previously given translational and rotation displacements, which are extrinsic parameters of the vision sensor. The objective function to optimize the translational and rotational displacements of different units is set as a normalized vector objective function, as follows [27]:

$$F_{penalty} = \underset{\hat{P}_{set}}{\arg\min} \sum_{i=1}^{N_{\max\_gen}} \frac{(\hat{D}_i - D_i)}{\max D_i - \min D_i} \tag{7}$$

where $\hat{D}_i$ and $D_i$ are the true and estimated displacements. The chromosome with the lowest penalty value has a higher probability of being selected in the next generation. The selected best chromosome is reproduced to form a new population, and crossover and mutation are performed to prevent GA from converging on local minima. Based on the updated population, Steps 2–4 are looped until the stopping criteria are satisfied, or the number of generations reaches the maximum number of generations. The parameter set with minimum penalty value is selected, and the constituted equation is automatically tuned. In this study, a single point crossover, proportional roulette wheel selection, and single point mutation method are used [28,29]. The population size of 150, percent probability of crossover of 0.6%, percent probability of mutation of 0.05%, and maximum number of generations of 200 are used.

---

**Algorithm 1.** Procedure of optimizing intrinsic parameters of the vision sensor with genetic algorithm.

---

*Input*:
    Population size, $n$
    Maximum number of iterations, $N_{\max\_gen}$
    Initial values and the searching area of the chromosomes, $P$
*Output*:
    Global best solution, $P_{bt}$

---

*begin*
    **Step 1**: Generate the initial population of chromosomes
    $P_{set} = [f_u, f_v, c_u, c_v, \mathbf{K}]$
    **while** satisfaction of stopping criteria **OR** number of generations is less than the
        maximum number of generations
        **Step 2**: Evaluate the penalty of each chromosome, $P_i$ ($I = 1, 2, \cdots, n$)
$$F_{penalty} = \underset{\hat{P}_{set}}{\mathrm{argmin}} \sum_{i=1}^{N_{\max\_gen}} \frac{(\hat{D}_i - D_i)}{\max D_i - \min D_i}$$
        **Step 3**: Select the best chromosome, and do reproduction
        **Step 4**: Perform the crossover and mutation
    **end**
    **Step 5**: Achieve the best individual in all generation, $P_{bt}$
*end*

---

To set the searching range of the parameters to be tuned, intrinsic parameters calculated by using checkerboards are analyzed, and the coefficient of variation, also called relative standard deviation, is calculated [30]. Figure 4 shows the checkerboards with different sizes. Table 1 shows the intrinsic parameters of each case with the combinations of one or two different sized checkerboards that are estimated. Figure 5 shows the box plots and coefficient of variations that are calculated. In this paper, the searching range of the parameters in the genetic algorithm is set from the calculated interquartile range in the box plots. Since the relative standard deviations of radial distortion parameter on the *Y* axis, and tangential distortion on the *X* and *Y* axes, show relatively large, the searching range is additionally multiplied by the weights on the three distortion parameters.

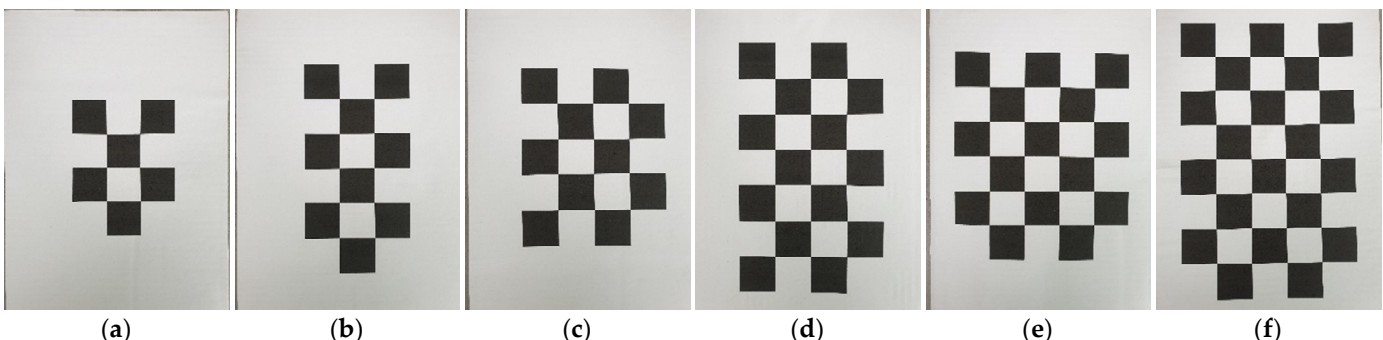

    **(a)**        **(b)**        **(c)**        **(d)**        **(e)**        **(f)**

**Figure 4.** Checkerboards with (**a**) 3 × 4, (**b**) 3 × 6, (**c**) 4 × 5, (**d**) 4 × 7, (**e**) 5 × 6, and (**f**) 5 × 8 squares for the estimation of the intrinsic parameters of the vision sensor.

**Table 1.** Estimated intrinsic parameters using different sets of checkerboards. The checkerboard '(a)' denotes the size and configuration of the checkerboard presented in Figure 4a.

| Checker-Boards | Focal Length | | Principal Points | | Radial Distortion | | Tangential Distortion | |
|---|---|---|---|---|---|---|---|---|
| | $F_x$ | $F_y$ | $C_x$ | $C_y$ | $K_1$ | $K_2$ | $K_3$ | $K_4$ |
| (a) | 2654.95 | 2667.25 | 692.02 | 421.61 | −0.3975 | 0.1285 | −0.0012 | 0.0130 |
| (a), (b) | 2634.64 | 2624.02 | 1023.62 | 380.04 | −0.4629 | 0.3087 | 0.0059 | 0.0008 |
| (a), (c) | 2578.67 | 2572.71 | 948.75 | 459.45 | −0.4461 | 0.1865 | 0.0017 | 0.0037 |
| (a), (d) | 2586.03 | 2584.54 | 1001.43 | 424.18 | −0.4436 | 0.2398 | 0.0002 | 0.0006 |
| (a), (e) | 2598.73 | 2582.47 | 990.96 | 431.47 | −0.4178 | 0.0585 | 0.0050 | 0.0029 |
| (a), (f) | 2363.73 | 2370.19 | 1044.13 | 347.82 | −0.4688 | 0.3937 | −0.0005 | −0.0020 |
| (b) | 2705.33 | 2673.96 | 1029.39 | 219.31 | −0.4917 | 0.5865 | 0.0155 | −0.0029 |
| (b), (c) | 2615.95 | 2614.07 | 997.06 | 411.56 | −0.4325 | 0.2560 | 0.0014 | 0.0037 |
| (b), (d) | 2734.70 | 2649.68 | 930.35 | 253.04 | −0.3618 | −1.0397 | 0.0278 | 0.0034 |
| (b), (e) | 2621.91 | 2601.25 | 981.15 | 414.57 | −0.4233 | 0.1851 | 0.0070 | 0.0043 |
| (b), (f) | 2645.72 | 2647.74 | 998.71 | 327.04 | −0.4519 | 0.3165 | 0.0025 | 0.0003 |
| (c) | 2665.99 | 2665.67 | 887.69 | 365.05 | −0.5030 | 0.9319 | 0.0060 | 0.0119 |
| (c), (d) | 2590.24 | 2598.32 | 921.28 | 451.59 | −0.4460 | 0.2329 | −0.0035 | 0.0065 |
| (c), (e) | 2611.21 | 2601.35 | 984.80 | 439.40 | −0.4289 | 0.1636 | 0.0026 | 0.0034 |
| (c), (f) | 2584.71 | 2574.96 | 1007.15 | 351.71 | −0.4498 | 0.0726 | 0.0016 | −0.0016 |
| (d) | 2716.70 | 2700.58 | 938.09 | 197.31 | −0.4994 | 0.4672 | 0.0163 | 0.0039 |
| (d), (e) | 2585.51 | 2583.79 | 1009.34 | 408.67 | −0.4050 | 0.1307 | 0.0002 | 0.0029 |
| (d), (f) | 2658.12 | 2629.06 | 1053.09 | 263.17 | −0.4885 | 0.2479 | 0.0145 | −0.0042 |
| (e) | 2552.71 | 2543.47 | 979.48 | 434.32 | −0.4422 | 0.5115 | −0.0010 | 0.0042 |
| (e), (f) | 2621.56 | 2604.26 | 994.28 | 321.01 | −0.4891 | 0.2585 | 0.0120 | −0.0048 |
| (f) | 2770.83 | 2755.42 | 1027.54 | 124.33 | −0.4865 | 0.5711 | 0.0137 | 0.0018 |

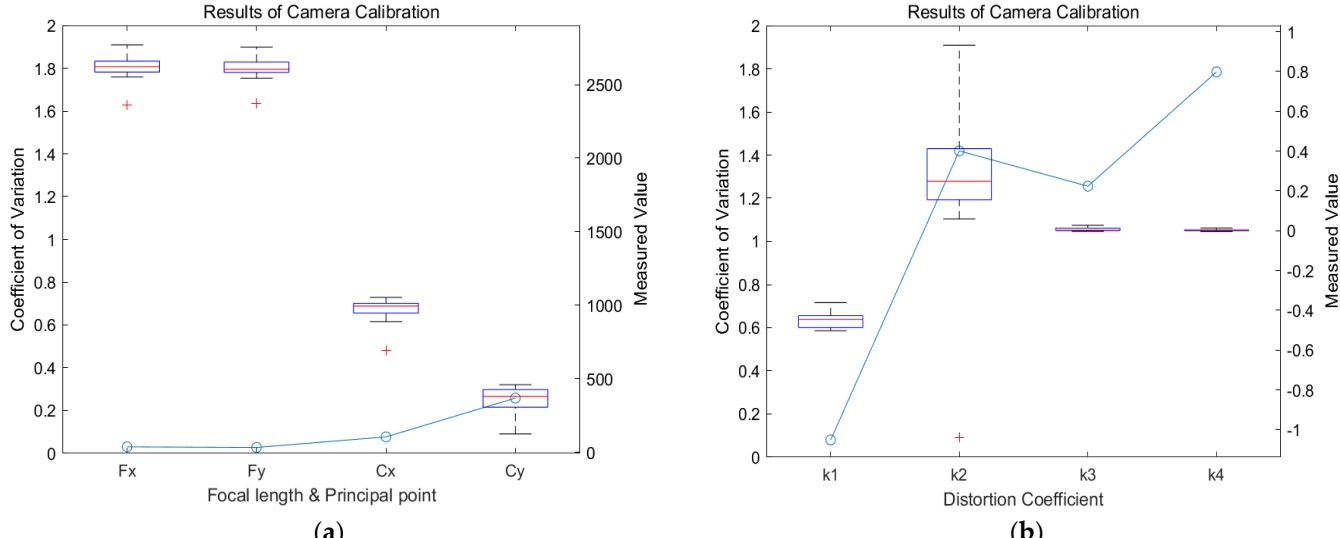

(**a**)　　　　　　　　　　　　　　(**b**)

**Figure 5.** Box plot ((**a**) parameters of the focal length and principal points, (**b**) parameters of the distortion) and coefficient of variation of the estimated intrinsic parameters of the vision sensor.

## 4. Experimental Tests

### 4.1. Verification of Displacement Estimation Using Model Structures

To verify the performance of the application of a genetic algorithm, experimental tests with artificial structures and a motion stage were performed. The structures were produced by simulating the shapes of actual port structures, and the relative displacement between the target structure placed on the motion stage and the fixed structure were estimated (see Figure 6). Figure 7 shows the graphic user interface based on visual c++, which employs image binarization using adaptive threshold, edge detection in subpixel level, and the camera extrinsic parameter estimation, which is developed to find the relative displacement

in a captured image. The estimated relative displacement between the two structures in the before and after images, the movement of the target structure according to the change of time, is calculated.

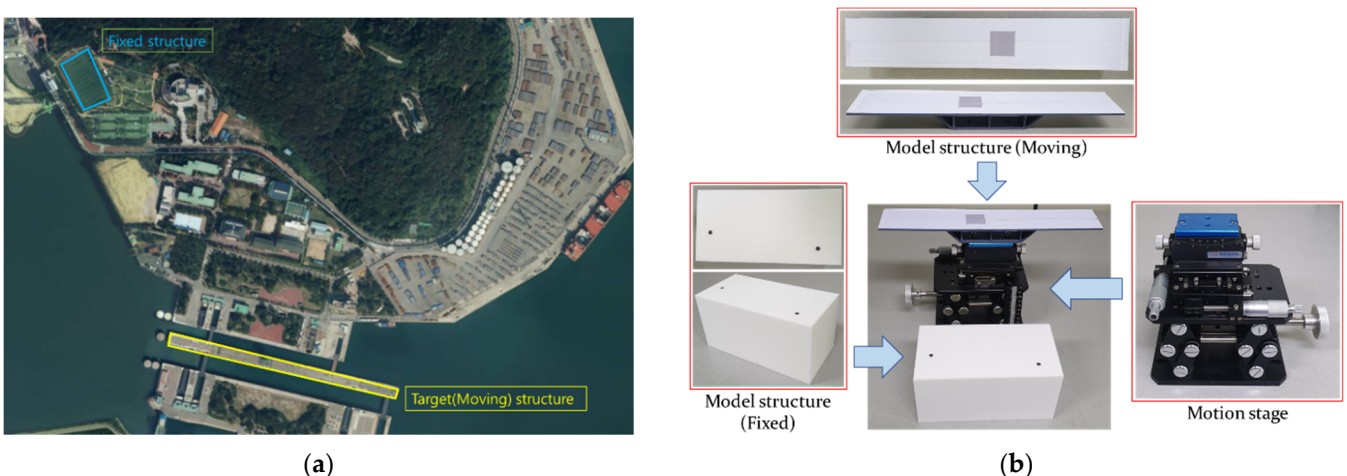

Figure 6. Experimental setup (**a**) port facilities for displacement measurement; (**b**) model structures and the motion stage.

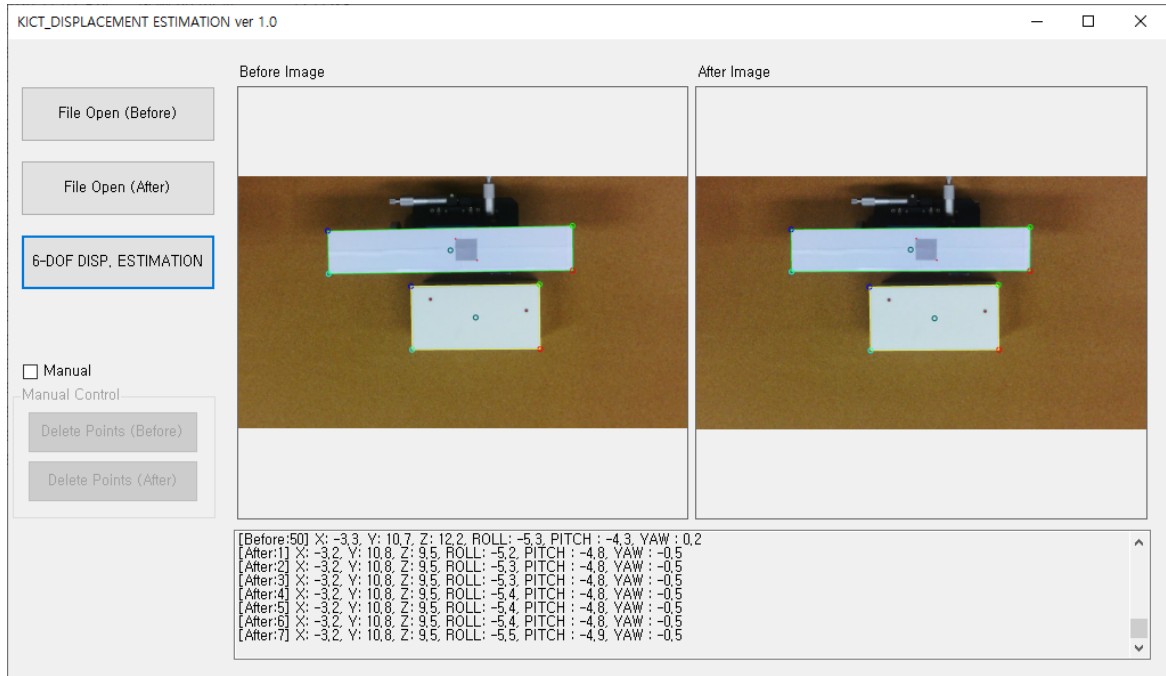

**Figure 7.** Graphic User Interface for estimating 6-DOF displacement.

By using different patterns and size of the checkerboards and experimental data sets with the *X*-axis translational displacement and *Y*-axis rotational displacement, intrinsic parameters are calculated (see Table 1). The median, minimum, and maximum values are used to generate populations of the chromosomes in GA. Since the relative standard deviations of radial distortion parameter on the *Y* axis, and tangential distortion on the *X* and *Y* axes show relatively large, as shown in Figure 5, the weights on the three distortion parameters are set to be 2.5 to enlarge the searching range. Table 2 shows the translational and rotational displacement results using the camera-intrinsic parameter adjusted by applying GA in the calculation of the 6-DOF displacement. The experimental test without GA has been performed with intrinsic parameters calculated by 40 captured images, using a checkerboard shown in Figure 4f. The table includes error of 6-DOF displacements

calculated based on ten different GA parameters and actual movement. The results show that the estimated displacements with the compensated camera-intrinsic parameters show better performance in both the translational and rotational displacements estimation. In the design standard for port and harbor structures [31–33], the maximum allowable horizontal displacement at the functional performance level is 100 mm. Considering the acceptable measurement tolerance, the proposed method with the RMSE of less than 3 mm and 1° for translational and rotation displacements, respectively, can be applied to the port structures to monitor the structural condition.

**Table 2.** Root mean square error (RMSE) of 6-DOF displacement estimation.

| Experimental Test | Translation | | | | | | Rotation | | | | | |
|---|---|---|---|---|---|---|---|---|---|---|---|---|
| | w/o GA | | | w/ GA | | | w/o GA | | | w/ GA | | |
| RMSE of Case 1 (−10mm translational movement along X-axis) | 0.8485 | | | 0.6122 (−28%) | | | 0.7703 | | | 0.3020 (−61%) | | |
| Errors of experimental results with ten different GA parameters — 1-1 | −1.0 | 0.4 | 1.0 | 0.4 | −0.1 | 0.5 | 0.0 | −1.3 | 0.3 | 0.2 | −0.6 | 0.0 |
| 1-2 | | | | 0.7 | 0.0 | 0.1 | | | | 0.2 | −0.4 | 0.0 |
| 1-3 | | | | 1.0 | −0.4 | 0.7 | | | | 0.3 | −0.2 | 0.0 |
| 1-4 | | | | 0.7 | −0.4 | 0.4 | | | | 0.3 | −0.3 | 0.0 |
| 1-5 | | | | −0.2 | −0.2 | 1.0 | | | | 0.3 | −1.0 | 0.0 |
| 1-6 | | | | 0.9 | −0.4 | 0.8 | | | | 0.3 | −0.4 | 0.0 |
| 1-7 | | | | 1.1 | −0.4 | 0.4 | | | | 0.3 | −0.1 | 0.0 |
| 1-8 | | | | 1.3 | −0.5 | 0.4 | | | | 0.4 | 0.0 | −0.1 |
| 1-9 | | | | 0.4 | −0.3 | 0.6 | | | | 0.3 | −0.5 | 0.0 |
| 1-10 | | | | 1.1 | −0.6 | 0.3 | | | | 0.5 | −0.1 | 0.0 |
| RMSE of Case 2 (5° rotational movement about Y-axis) | 2.3188 | | | 2.0538 (−11%) | | | 0.8794 | | | 0.5255 (−40%) | | |
| Errors of experimental results with ten different GA parameters — 2-1 | 2.7 | −1.0 | −2.8 | 2.8 | 0.0 | −2.5 | 0.0 | −1.4 | 0.6 | −0.8 | −0.7 | 0.0 |
| 2-2 | | | | 3.1 | 0.0 | −0.1 | | | | −0.6 | −0.7 | −0.1 |
| 2-3 | | | | 3.1 | −0.2 | −1.2 | | | | −0.7 | −0.6 | 0.0 |
| 2-4 | | | | 2.3 | 0.1 | −2.8 | | | | −0.9 | −1.0 | 0.0 |
| 2-5 | | | | 3.0 | 0.4 | 0.5 | | | | −0.8 | −0.8 | −0.2 |
| 2-6 | | | | 3.4 | −0.3 | −1.2 | | | | −0.6 | −0.4 | 0.0 |
| 2-7 | | | | 3.1 | −0.2 | −1.5 | | | | −0.6 | −0.5 | 0.0 |
| 2-8 | | | | 3.9 | −0.8 | −1.7 | | | | −0.4 | −0.1 | 0.0 |
| 2-9 | | | | 3.2 | −0.4 | −1.0 | | | | −0.4 | −0.5 | 0.0 |
| 2-10 | | | | 2.7 | −0.5 | −2.8 | | | | −0.7 | −0.9 | 0.0 |
| RMSE of Case 3 (−30 mm translational movement along Z-axis) | 3.4113 | | | 2.3104 (−32%) | | | 0.6683 | | | 0.4607 (−31%) | | |
| Errors of experimental results with ten different GA parameters — 3-1 | 1.9 | 4.9 | 2.7 | 1.7 | 1.8 | −4.1 | 1.1 | 0.2 | −0.3 | 0.7 | −0.1 | −0.2 |
| 3-2 | | | | 2.5 | 1.3 | −3.7 | | | | 0.6 | −0.1 | −0.2 |
| 3-3 | | | | 1.9 | 1.9 | −2.2 | | | | 0.8 | −0.4 | −0.2 |
| 3-4 | | | | 1.3 | 1.9 | −3.4 | | | | 1.0 | −0.1 | −0.2 |
| 3-5 | | | | 2.7 | 1.6 | −2.8 | | | | 0.4 | −0.4 | −0.2 |
| 3-6 | | | | 2.1 | 2.1 | −2.8 | | | | 0.8 | 0.0 | −0.2 |
| 3-7 | | | | 1.9 | 2.0 | −3.2 | | | | 0.8 | 0.0 | −0.2 |
| 3-8 | | | | 1.5 | 2.1 | −2.4 | | | | 0.7 | −0.3 | −0.2 |
| 3-9 | | | | 1.7 | 2.1 | −2.2 | | | | 0.7 | −0.2 | −0.2 |
| 3-10 | | | | 1.0 | 2.2 | −2.5 | | | | 0.6 | −0.6 | −0.2 |

### 4.2. Verification of Field Applicability Using Port Structure Images

To verify the applicability of the proposed method, an experimental test with an image of one of the major port structures in the Incheon Republic of Korea was performed. An inspection drone specialized for port facilities was developed containing the following: a module for precise three-dimensional position control using multiple GNSS and corrected

signals, a module for mounting a multi-angle camera and a front gimbal, and a folding frame capable of being carried by a person for photo and videography (see Figure 8a). The Figure 8b shows the 3D flight trajectory when capturing the images at high altitude. Through the development of real-time image streaming control technology that integrates the ground control module and the LTE module, it is possible to control the drone in the invisible area more than 3 km away from Incheon port.

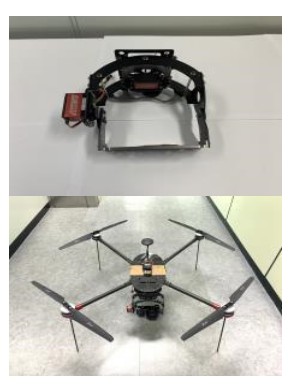
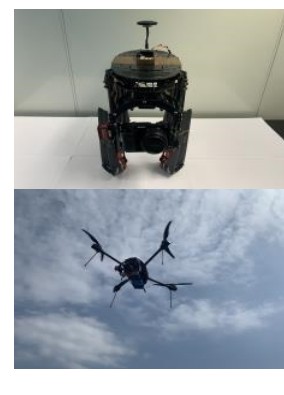
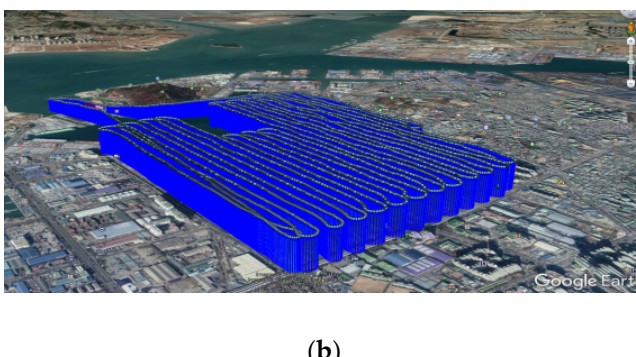

(**a**)                                                                                              (**b**)

**Figure 8.** Experimental setup with (**a**) a drone with a front gimbal and its (**b**) flight trajectories at Incheon port.

The artificial movement of the structure was generated by moving the target structure using an integrated orthophoto, and the relative displacement of the structure between two images was calculated as shown in Figure 9. The figure shows that the main displacement is predicted by the *X*-axis displacement, which is the longitudinal directions of the target structure. The estimated relative displacement in the test is found to be $D = [-43{,}011.9, -825.5, 439.6, -1.2, 0, 2.6]$ with all units in mm or degrees. The intrinsic parameters were tuned by using GA with the specifications of the structures, which are the coordinates of feature points in the fixed and the target structures. By using the proposed method, it will be possible to determine whether to continue using the port structures by estimating the displacement before and after a disaster.

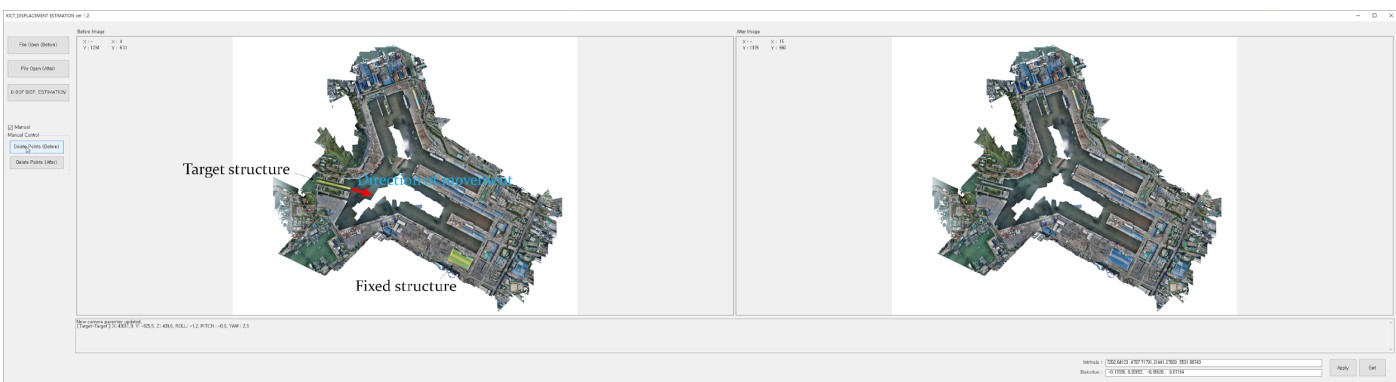

**Figure 9.** Estimation of the relative displacement using port structure images.

## 5. Conclusions

The translational and rotational displacements of port structures can be estimated by capturing images that include both a fixed and a target structure. The movement of the target structure relative to the fixed structure can be calculated by estimating the displacements from the camera to the fixed and target structures, respectively. The movement of the structure can be measured by the vision sensor mounted on mobile platforms such as drones without attaching a special sensing system to the structure. Genetic algorithm was introduced to improve the accuracy of the displacements, and the results confirmed

that the root mean square errors of translational and rotational displacement were greatly reduced. The applicability of the proposed method to port infrastructure was verified using high-latitude orthogonal images, and the specifications of the structures with a mobile platform. In the future, deep learning techniques will be applied to enable robust detection of the structures against changes in external environmental conditions and ensure usability and safety of constantly monitored major port facilities.

**Author Contributions:** H.J. conceived the presented idea and supervised the project. J.M., H.B. and Y.B. developed the detection and quantification method and performed the experimental tests. All authors have read and agreed to the published version of the manuscript.

**Funding:** This research was a part of the project titled 'Development of smart maintenance monitoring techniques to prepare for disaster and deterioration of port infra structures (No. 20210659)' funded by the Ministry of Oceans and Fisheries, Korea.

**Institutional Review Board Statement:** Not applicable.

**Informed Consent Statement:** Not applicable.

**Data Availability Statement:** Data available on request due to restrictions e.g., privacy or ethical.

**Acknowledgments:** This research was a part of the project titled 'Development of smart maintenance monitoring techniques to prepare for disaster and deterioration of port infra structures (No. 20210659)' funded by the Ministry of Oceans and Fisheries, Korea. The port images obtained by the drone in Figure 8 were provided by SISTECH, Korea.

**Conflicts of Interest:** The authors declare that they have no conflict of interest.

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
