# Peer review of "Port Structure Inspection Based on 6-DOF Displacement Estimation Combined with Homography Formulation and Genetic Algorithm"

_applsci, doi:10.3390/app11146470_

Round 1

Reviewer 1 Report

This paper proposes a vision sensor based 6-DOF displacement evaluation method incorporating a genetic algorithm.  The proposed approach is used to monitor the critical defects of port infrastructure, such as deflection, slope, and slip. A genetic algorithm is used to adjust the intrinsic parameters.

Overall, it is good research, and the structure is good. Some detailed suggestions are as follows to further improve it.

  1. A paragraph to outline the remaining parts of the paper is recommended to be added at the end of the Introduction part.
  2. There are many algorithms, why do you choose GA? The reason of utilizing GA should be enhanced.
  3. In Section 3, more details about GA and how does it optimize the camera intrinsic parameters should be provided.
  4. Literature review on existing heuristics algorithms (e.g., differential evolution, particle swarm optimization) should be extended. References of some popular algorithms is helpful. (1) Comprehensive learning particle swarm optimization algorithm with local search for multimodal functions. IEEE Transactions on Evolutionary Computation 23 (4), 718-731, 2019. (2) Truss topology, shape and sizing optimization by fully stressed design based on hybrid grey wolf optimization and adaptive differential evolution. Engineering Optimization, 50(10), 1645-1661, 2018, etc.
  5. In Section 4, key parameters for the tests should be provided. How many independent runs have been done by GA? Statistical results from multiple runs of a heuristic algorithm are expected.
  6. In Figure 3, circles are in different colors. Do they have special meanings?

Reviewer 2 Report

  1. This paper describes a vision sensor based 6-DOF displacement evaluation method for port structure. The method calculates the relative position of the target port structure and measures the movement of the structure over time. To my knowledge, the measures technology applied in this manuscript is not a novel method, but the application field is new and potentially.
  2. In table 3, the result of the genetic algorithm seems obvious. But there has only the root mean square errors. Could authors provide the error between the average measurement result and the actual movement? It will be a help to evaluate the advantages of this proposed method.
  3. Could the authors show the comparison between the generally acceptable measurement tolerance of the port structure and the measurement accuracy of the method in this manuscript? Otherwise, the reader cannot understand the applicability of the proposed method.

Author Response

This manuscript is a resubmission of an earlier submission. The following is a list of the peer review reports and author responses from that submission.

Round 1

Reviewer 1 Report

This paper presents a visual technique for computing the rigid body motion (in all six degrees of freedom) of a body, and applies this technique to port structures. It's unclear if the method is applicable to deformable bodies, and thus I'm not sure if this can truly be used to monitor these port structures (I admit, I do not know the stiffness of the structures that the authors are proposing to use this method on). The idea is interesting, but not well explained.

Validation is practically not done. The laboratory experiment is not well explained, so it is not clear to what the GA method is being compared. The field applicability is not even tested in the field... it is tested on a photo that is edited to mimic a certain amount of displacement. 

References 8-16, which appear to be the only references cited on this topic are all from what seems to be the same research group including the last author, Haemin Jeon. Please include other references on non-contact displacement measurement.

Page 2 - Please define the camera's intrinsic parameter and homography matrix in this introduction. It is not at all clear what these represent.

Equation (3) - Matrix dimensions do not agree. I do not think there should be a "1" in the [Xc, Yc, Zc] matrix.

Page 3 - What does it mean that "cu and cv represent the principal points calculated a priori using camera calibration technique."  This is not clear.

Page 3 - Definition of y seems incorrect. I believe y = Yc/Zc, not Xc/Zc?

Page 3 - Final paragraph on page 3 does not explain exactly how the displacements are computed using Equations (1) through (3) or why 4 or more points are needed to do this. It merely states that "displacement between the camera and the structures can be estimated."

Page 4 - Equations (4) and (5) are not clear. Does the dot (multiplication) represent a matrix multiplication, a dot product, or an element by element multiplication? I also believe Equation (5) has an extra "t" in the superscript.

Page 4 - While it may be theoretically true that the genetic algorithm (GA) will guarantee convergence to the global optimum given infinite generations and an enormous population size, in practice this is difficult to achieve or assess. How are the authors certain that their application of GA is not converging to some local minima here?

Page 4 - Please define the K vector. I assume this is [K1 K2 K3 K4]?

Page 5 and Table 2 - Please describe how these checkerboards are being used to compute the intrinsic parameters. Might these parameters change for different applications, air conditions, distance to target, etc., or are they just properties of the camera? 

Figure 5 - There are small red plus-signs in both box plots that are unlabeled. I have no idea what these mean. Furthermore, the distortion coefficients K3 and K4 only have a large coefficient of variation because these parameters are so close to zero. In reality, it seems these are parameters are fairly well known.

Table 3 - This table compares the errors in estimating the displacements with and without using GA optimization. However, there is no discussion for what intrinsic parameters were used in the analysis without GA. Second, exactly what was used to optimize the GA in this case? Was the GA optimized on some other independent experiment, and then applied to this experiment, or was the GA optimized to fit the known displacement values of this experiment? If the latter is true, then this is not proper or useful validation that the GA works, because an optimized GA will, by definition, perform better than an unoptimized solution - that is trivial.

Page 8 - Virtually no information was given on the use of this method for analyzing an actual port structure. In fact, no such analysis was done. Instead, the method is tested on a modified (doctored) photo that simulates displacement. This proves nothing about how this might work in a true field application. Furthermore, no information is given as to how the GA was used to optimize the intrinsic parameters - doesn't this optimization require knowledge of the true displacements?  

Reviewer 2 Report

This paper proposes a vision sensor based 6-DOF displacement evaluation method incorporating a genetic algorithm.  The proposed approach is used to monitor the critical defects of port infrastructure, such as deflection, slope, and slip. A genetic algorithm is used to adjust the intrinsic parameters.

Overall, it is good research, and the structure is good. Some detailed suggestions are as follows to further improve it.

  1. A paragraph to outline the remaining parts of the paper is recommended to be added at the end of the Introduction part.
  2. There are many algorithms, why do you choose GA? The reason of utilizing GA should be enhanced.
  3. In Section 3, more details about GA and how does it optimize the camera intrinsic parameters should be provided.
  4. Literature review on existing heuristics algorithms (e.g., differential evolution, particle swarm optimization) should be extended. References of some popular algorithms is helpful. (1) Comprehensive learning particle swarm optimization algorithm with local search for multimodal functions. IEEE Transactions on Evolutionary Computation 23 (4), 718-731, 2019. (2) Truss topology, shape and sizing optimization by fully stressed design based on hybrid grey wolf optimization and adaptive differential evolution. Engineering Optimization50(10), 1645-1661, 2018, etc.
  5. In Section 4, key parameters for the tests should be provided. How many independent runs have been done by GA? Statistical results from multiple runs of a heuristic algorithm are expected.
  6. In Figure 3, circles are in different colors. Do they have special meanings?